# A Selective ALDH1A3 Inhibitor Impairs Mesothelioma 3-D Multicellular Spheroid Growth and Neutrophil Recruitment

**DOI:** 10.3390/ijms24076689

**Published:** 2023-04-03

**Authors:** Sara Boumya, Silvia Fallarini, Sonia Siragusa, Giovanni Petrarolo, Silvio Aprile, Valentina Audrito, Concettina La Motta, Silvia Garavaglia, Laura Moro, Giulia Pinton

**Affiliations:** 1Department of Pharmaceutical Sciences, University of Piemonte Orientale, 28100 Novara, Italy; sara.boumya@uniupo.it (S.B.); silvia.fallarini@uniupo.it (S.F.); sonia.siragusa@uniupo.it (S.S.); silvio.aprile@uniupo.it (S.A.); laura.moro@uniupo.it (L.M.); 2Department of Pharmacy, University of Pisa, 56100 Pisa, Italy; giovanni.petrarolo@phd.unipi.it (G.P.); concettina.lamotta@uniupi.it (C.L.M.); 3Department of Science and Technological Innovation, University of Piemonte Orientale, 15121 Alessandria, Italy; valentina.audrito@uniupo.it

**Keywords:** malignant pleural mesothelioma, ALDH1A3, *CDKN2A*, tumour-associated neutrophils

## Abstract

Aldehyde dehydrogenase 1A3 (ALDH1A3), one of the three members of the aldehyde dehydrogenase 1A subfamily, has been associated with increased progression and drug resistance in various types of solid tumours. Recently, it has been reported that high ALDH1A3 expression is prognostic of poor survival in patients with malignant pleural mesothelioma (MPM), an asbestos-associated chemoresistant cancer. We treated MPM cells, cultured as multicellular spheroids, with NR6, a potent and highly selective ALDH1A3 inhibitor. Here we report that NR6 treatment caused the accumulation of toxic aldehydes, induced DNA damage, *CDKN2A* expression and cell growth arrest. We observed that, in *CDKN2A* proficient cells, NR6 treatment induced *IL6* expression, but abolished *CXCL8* expression and IL-8 release, preventing both neutrophil recruitment and generation of neutrophil extracellular traps (NETs). Furthermore, we demonstrate that in response to ALDH1A3 inhibition, *CDKN2A* loss skewed cell fate from senescence to apoptosis. Dissecting the role of ALDH1A3 isoform in MPM cells and tumour microenvironment can open new fronts in the treatment of this cancer.

## 1. Introduction

Malignant pleural mesothelioma (MPM) is a deadly cancer caused by exposure to asbestos fibers, which originates from mesothelial lining cells of the pleura. MPM is characterized by a long latency period and a life expectancy of 11–12 months after diagnosis [1,2,3,4]. For several decades, cisplatin plus pemetrexed-based chemotherapy has been the approved first-line therapy for MPM patients, with no novel treatments demonstrating superior response rate [5]. In 2021, the publication of results from the phase III CheckMate 743 trial [6] showed that the combination of nivolumab plus ipilimumab had a favourable clinical benefit–risk profile and gained approval from the regulatory agencies worldwide as a new option for patients with unresectable MPM. Extended follow-up, as well as additional analyses of candidate biomarkers of immunotherapy efficacy remain of continued interest and require further investigation.

Recently, it has been reported that *CDKN2A* not only serves as a diagnostic marker, but may determine cell fate in response to EZH2 (Enhancer of zeste homolog 2) or immune checkpoint inhibition [7,8]. Homozygous deletion of *CDKN2A* (which encodes for the cell cycle inhibitor p16^ink4a^), along with BAP1 inactivation, represents the most common genetic aberration in MPM and is associated with poor outcome for MPM patients [9]. The work with abemaciclib confirms the importance of *CDKN2A* deletion as therapeutic target in MPM [10].

Despite scientific advances, treatment options for MPM remain limited, with no approved second line therapies available. Therefore, continued research focused on identifying new and promising targeted therapies is crucial. In this context, it has been reported that the subfamily of aldehyde dehydrogenase 1A (ALDH1A), that belongs to the aldehyde dehydrogenase (ALDH) superfamily of 19 isoenzymes, play a key role in the progression, maintenance and drug resistance of various tumours [11].

ALDHs exert a protective role in cellular defence against oxidative stress and reactive oxygen species (ROS), through a NAD (P)^+^ dependent reaction of oxidation of the carbonyl groups to the corresponding carboxylic acid, to avoid the accumulation of toxic aldehydes such as acrolein, malondialdehyde (MDA), 4-hydroxy-2-nonenal (4-HNE) and 4-hydroxy-2-hexanal (4-HHE) [12]. Endogenous aldehydes are generated during the metabolism of amino acids, alcohols, lipids, and vitamins, while exogenous aldehydes, as intermediates or products, are derived from the metabolism of a wide variety of environmental agents and drugs [13]. Beyond their main role in the detoxification of aldehydes, ALDHs have also an essential role in the biosynthesis of key metabolic regulators of cellular homeostasis, such as retinoic acid (RA), γ-aminobutyric acid (GABA), dopamine, and betaine [14].

ALDH1A subfamily is composed by the three different isoforms: ALDH1A1, ALDH1A2 and ALDH1A3. Despite their high similarity, the three isoforms exhibit preference for specific substrates or biological endpoints, and their expression patterns do not entirely overlap [15]. Among the three isoforms, ALDH1A3 has been described to be overexpressed in different neoplasms, including pancreatic cancer, high-grade gliomas, and ovarian cancer, but is not expressed in the non-neoplastic cells. Specifically, ALDH1A3 is highly expressed in drug resistant cancer stem cells (CSCs), characterized by the capability to promote self-renewal, clonogenic growth and metastases [16,17]. Recently, an analysis of 84 MPM patients from the TGCA database revealed that high ALDH1A3 expression is significantly associated with worse prognosis [18]. Canino et al. demonstrated that in MPM derived cell lines and in primary cells the platinum-based treatment triggered the emergence of strongly chemoresistant cell subpopulations exhibiting high levels of ALDHs enzymatic activity; ALDH1A3 was identified as the main contributor [19].

ALDH1A3 represents a very promising and intriguing therapeutic target, but the development of selective inhibitors for each ALDH1A isoform has been hampered by a high degree of sequence and structural homology. Very few compounds have been described as selective for ALDH1A3 enzyme [20]. MCI-INI-3 is a potent selective inhibitor of recombinant human ALDH1A3, with greater than 140-fold selectivity for ALDH1A3 compared to the isoform ALDH1A1 [21]. YD1701 showed stronger binding to ALDH1A3 than other ALDH isoforms [22].

In this context, NR6 has been recently proposed as a novel, highly selective ALDH1A3 inhibitor [23,24,25,26]. The functional profile of NR6, from the biochemical, cellular and structural points of view, highlights that it is a potent and selective inhibitor (IC_50_ = 5.3 ± 1.5 μM and K_i_ = 3.7 ± 0.4 μM). A close inspection of the NR6-ALDH1A3 crystallographic structure revealed that NR6 binds the tyrosine residue 472 (Y472) of ALDH1A3, which is non-conserved in all the other human ALDH1A isoenzymes, and this drives its selectivity against this isoform. The Y472 residue is essential in coordinating with the inhibitor because it differentiates the binding pocket from that of the parent isoenzymes [23,24,25,26].

MPM is a highly complex tumour, not only to treat but also to understand, due to the involvement of different players in the tumour organization and in the principal pro-tumoural functions [27,28]. The MPM immune microenvironment is unique and complex, characterized by great intra- and inter-tumoural heterogeneity.

Neutrophils have been extensively described in the pathophysiology of autoimmune and infectious diseases, but increasing evidence suggests their important role in cancer progression, through their interaction with tumour and immune cells in the blood and in the tumour microenvironment (TME) [29]. The blood neutrophil-to-lymphocyte ratio (NLR) has been studied in several solid tumours to predict survival and response to cancer therapies. As suggested by different studies, NLR represents an independent predictor of survival also for MPM patients as well. The hypothesis behind this is that activated T cells might be suppressed by marked neutrophil infiltration. In fact, a high NLR could decrease the effects of the lymphocyte-mediated cellular immune response and promote tumour progression/maintenance [30,31]. Recently, the role of neutrophils in cancer has attracted attention because they are heterogeneous in phenotype and function and can display outstanding plasticity, depending on the context, exerting anti- or pro-tumourigenic functions. As is well known, neutrophils, after their mobilization from the bone marrow, are recruited to the tumour site by the action of neutrophil-attracting chemokines, mainly IL-8, that can be produced not only by other immune cells but also directly by tumour cells and cancer associated fibroblasts [32]. Once in the tumour tissue, the microenvironment cues are responsible for the polarization of tumour associated neutrophil (TAN) towards either anti-tumoural or pro-tumoural phenotypes [33]. Different signals that participate in TAN polarization have been identified. In particular, TGF-β has been found to polarize neutrophil functions in a pro-tumour direction characterized by high expression of ARG1, PD-L1, CCL7 and CXCL14 [34]. In contrast, IFN and GM-CSF drive neutrophils toward an anti-tumour state characterized by expression of MHCII and co-stimulatory molecule [35]. Anti-tumoural neutrophils can directly kill tumour cells, support T cell recruitment and antitumour activity, and suppress T-reg cell differentiation [36]. Through a sequence of processes defined as NETosis, activated neutrophils release NETs (Neutrophil Extracellular Traps), comprising decondensed chromatin (histones and DNA), into their surrounding matrix, forming three-dimensional protein structures with associated cytotoxic enzymes. Increasing evidence has reported that NETs have been directly associated with the initiation and induction of tumour invasion, linked to metastasis recurrence [37].

In this article, we describe the expression and role of ALDH1A3 in different MPM cell lines cultured as multicellular spheroids (MCSs). We report that NR6 treatment strongly reduced the proliferation rate of MPM MCSs and impaired neutrophil recruitment. In this scenario, *CDKN2A* tips the balance between apoptosis and senescence.

## 2. Results

### 2.1. MPM Cells, Cultured as Multicellular Spheroids, Express Different Levels of ALDH1A3

We evaluated the expression of ALDH1A3 in multicellular spheroids (MCSs) from REN, MSTO-211H, and H2596 cell lines derived from epithelioid, biphasic, and sarcomatoid MPMs, respectively. Representative light microscope images of the three cell lines grown as MCSs for 24, 48, and 72 h are shown in Figure 1A. At 72 h, MCSs were collected and mRNA and proteins were extracted. Real-time PCR analysis demonstrated that ALDH1A3 transcripts were expressed in H2596 and at higher levels in MSTO-211H, but not in REN cells (Figure 1B) and NP2 mesothelial cells (Appendix A). Western blot analysis confirmed that ALDH1A3 protein levels correlated well with the corresponding mRNA levels (Figure 1C).

### 2.2. ALDH1A3 Inhibition Causes Accumulation of Malondialdehyde and Induces a Senescence Growth Arrest in MSTO-211H MCSs

To evaluate the effects of ALDH1A3 inhibition, we treated MSTO-211H MCSs with 1 μM of the selective ALDH1A3 inhibitor, NR6. Light microscope images of untreated and NR6-treated (24 and 72 h) MSTO-211H MCSs are shown in Figure 2A. After 72 h of incubation, MCSs were dissociated, and the number of viable cells was counted. The graph in Figure 2B shows a significant reduction in the number of viable cells in MSTO-211H MCSs treated with NR6. NR6-treated REN and NP2, ALDH1A3 negative, cells exhibited no significant viability effects (Appendix A). As shown in Figure 2C, NR6 treatment caused in MSTO-211H MCSs the intracellular accumulation of malondialdheyde (MDA), whose nanomolar levels were determined after derivatization with dimedone by liquid chromatography coupled to high resolution mass spectrometry (LC-HRMS) analysis (Appendix A) [38].

Western blot analysis revealed increased H2AX phosphorylation (γ-H2AX) in NR6-treated MCSs, suggestive of increased DNA damage (Figure 2D). The graph in Figure 2E shows the reduced levels of total NAD (NADt), NADH, and NAD^+^ in NR6-treated MSTO-211H MCSs, compared to controls. As expected, treatment with NR6 significantly reduced the level of NADH, a product of ALDH1A3 activity. In addition, we observed a reduction in NAD^+^ levels, probably due to increased consumption by other NAD-consuming enzymes and/or conversion in NAD(P)^+^ to support antioxidant pathways. The ratio between NAD^+^ and NADH in control and NR6-treated MSTO-211H MCSs is reported in Figure 2F.

The reduction in cell number, observed upon NR6 treatment, was not due to apoptotic death as demonstrated by no PARP-1 cleavage (Figure 2G) and induced expression of *BCL2L11* (encoding Bim) and *BBC3* (encoding PUMA) (as shown in the following Section 2.4). Conversely, NR6 treatment induced the expression of *CDKN2A* (coding for the cell cycle regulator p16^ink4a^) and *IL6* suggestive of a secretory senescent phenotype (Figure 2H). In NR6-treated MCSs, differently from *IL6*, *CXCL8* expression was significantly reduced (Figure 2H).

### 2.3. ALDH1A3 Inhibition Counteracts IL-8 Secretion and Neutrophil Recruitment in MSTO-211H MCSs

In accordance with reduced *CXCL8* transcripts (Figure 2H), we found that the IL-8 levels released in the medium by NR6-treated MCSs were significantly lower than controls (Figure 3A). As IL-8 is one of the main cytokine chemoattractants for neutrophils, we evaluated the capability of MCSs to recruit naïve neutrophils in a co-culture model. After 48 h NR6 treatment of MSTO-211H MCSs, 4 × 10^4^ neutrophils, isolated from healthy donors, were added to each MCS and incubated for additional 24 h. Figure 3B shows representative light microscope images of MCSs untreated or treated with NR6 in co-culture with neutrophils. NR6 treatment significantly reduced neutrophils infiltration in MSTO-211H MCSs. To exclude a direct effect of NR6 on neutrophils, we assayed ALDH1A3 expression by Western blot analysis and performed a cell viability assay upon NR6 treatment. As shown in Figure 3C, neutrophils isolated from a pool of three healthy donors did not express ALDH1A3 and their viability was not affected by NR6 treatment (Figure 3D). By confocal microscopy analysis with an anti-CD66b-FITC antibody, we confirmed a reduction in neutrophils infiltration in NR6-treated MCSs compared to controls (Figure 3E). 4′,6-diamidin-2-fenilindolo (DAPI) was used as a nuclear counterstain. Data were confirmed evaluating, by citofluorimetric analysis, the percentage of CD66b-FITC positive neutrophils in dissociated MCSs (Figure 3F). Interestingly, as evidenced by confocal microscopy after immunostaining with antibodies specific for myeloperoxidase (MPO-green) and citrullinated Histone H3 (citH3-red), neutrophils that penetrated the untreated MSTO-211H MCSs underwent NETosis (Figure 3G).

### 2.4. ALDH1A3 Inhibition Induces Apoptosis in CDKN2A Silenced MSTO-211H MCS

Homozygous deletion of the 9p21 locus, which encompasses *CDKN2A*, is frequent in MPM. We hypothesized that *CDKN2A* induction in wild type or hemizygous deleted cells, could be responsible for the arrest in cell growth observed in NR6-treated MSTO-211H MCSs. We generated MCSs from *CDKN2A* silenced MSTO-211H cells to counteract its induction mediated by NR6. As shown in bright field microscopy images (Figure 4A) and in graph in Figure 4B, NR6 treatment reduced the growth of cells transfected with non-specific siRNAs and more significantly of those having silenced *CDKN2A*. NR6 treatment induced H2AX phosphorylation (γ-H2AX) in both MCSs from cells transfected with non-specific (NS) and *CDKN2A* siRNAs (Figure 4C). Differently from NS transfected cells, in *CDKN2A* silenced cells, we observed induction of apoptosis, as demonstrated by bright field microscope images (Figure 4A), PARP-1 cleavage (Figure 4D) and the induction of *BCL2L11*, *BBC3* (Figure 4E). *CDKN2A* silencing was confirmed by real-time PCR analysis (Figure 4E). Furthermore, in *CDKN2A* silenced cells neither *IL6* nor *CXCL8* were induced by NR6 treatment (Figure 4E). Consistent with reduced *CXCL8* transcripts, and IL-8 release (Figure 4F), we observed reduced recruitment of neutrophils in both NS and *CDKN2A* silenced cells treated with NR6 (Figure 4G).

### 2.5. ALDH1A3 Inhibition Induces Apoptosis in CDKN2A Homozygous Deleted H2596 Cells Cultured as MCSs

To confirm data obtained by *CDKN2A* silencing in MSTO-211H cells, we treated MCSs generated from the *CDKN2A* homozygous deleted H2596 cell line with NR6, for 72 h. As observed in *CDKN2A* silenced MSTO-211H MCSs, we observed a reduction in size (Figure 5A) and cell number in H2596 MCSs treated with NR6 (Figure 5B). ALDH1A3 inhibition caused an increase in intracellular MDA level (Figure 5C) and γ-H2AX (Figure 5D) expression, suggestive of DNA damage. NR6 treatment induced apoptosis in H2596 MCSs, as demonstrated by bright field microscope images (Figure 5A), PARP-1 cleavage (Figure 5E) and induction of *BCL2L11*, *BBC3* (Figure 5F). As observed in *CDKN2A* silenced MSTO-211H, *IL6* and *CXCL8* were not induced by NR6 treatment of H2596 MCSs (Figure 5F). Consistent with reduced IL-8 release (Figure 5G), we observed a reduction in the recruitment of neutrophils in NR6-treated H2596 MCSs, compared with controls (Figure 5H).

## 3. Discussion

Malignant pleural mesothelioma (MPM) is considered a highly lethal condition due to its recurrence despite standard approaches [1]. Currently, there is no approved therapy for relapsed MPM after front-line treatment. The identification of molecular targets and small molecules as candidate targeted therapies for patients with MPM is urgently needed.

ALDH1A3 expression has been associated with worse survival outcomes in a variety of cancers [39,40,41,42,43]. Recently, Cioce M. et al. reported that MPM patients with high ALDH1A3 expression levels had a poorer prognosis [18]. Further the prognostic potential, authors demonstrated that ALDH1A3 expression was responsible for the survival of MPM chemoresistant cell subpopulations. Indeed, the downregulation of ALDH1A3 expression in MPM cells increased sensitivity to pemetrexed and cisplatin [19].

Here we describe that treatment with the highly selective ALDH1A3 inhibitor, NR6, impaired the growth of ALDH1A3 positive MPM cells cultured as multicellular spheroids (MCSs). Treatment of MPM MCSs with NR6 caused the intracellular accumulation of malondialdehyde (MDA) and DNA damage.

Accumulation of lipid aldehydes, including MDA, results in DNA and protein adducts that lead to alterations in gene expression and protein activity and contribute to a persistent condition of cell stress damage. This stressed condition and persistence of DNA damage play a role in maintaining a senescent arrest [44]. ALDHs, including ALDH1A3, oxidize aldehydes to their corresponding acids, reactions that are coupled to the reduction in NAD^+^ to NADH [45]. Consistently, we observed a significant reduction in NADH levels resulting in an increased NAD^+^/NADH ratio in MPM MCSs treated with NR6. The NAD^+^/NADH balance is critical for maintaining redox homeostasis in cells and for influencing cellular energy metabolism by affecting the activity of NAD^+^-dependent enzymes, including PARPs, deacetylase sirtuins and several dehydrogenases involved in glycolysis and mitochondrial oxidative phosphorylation [46,47]. A global reduction in total NAD could reflect a deregulation of the above cited enzymes, of NAD biosynthetic pathways [46] and a global metabolic reprogramming in MCSs under the pressure of NR6. Another possibility could be a conversion of NAD^+^ in NAD(P)^+^, used in detoxification pathways [48] to contrast the cell stress damage induced by NR6. Further studies could clarify these hypotheses, with to the goal of identifying novel agents, including NR6, capable of affecting the NAD^+^/NADH ratio under pathological settings to achieve therapeutic effects, as demonstrated for example for KP372-1 as a potent NQO1-mediated redox cycling agent.

Furthermore, NR6 triggered *CDKN2A* expression in MSTO-211H cells, leading to sustained senescent arrest. Despite many years of research, cell senescence remains a somewhat enigmatic cell state. Among senescence-related mechanisms, the senescence-associated secretory phenotype (SASP) has gained considerable attention. Indeed, senescent cells secrete a variety of soluble molecules [49], including inflammatory cytokines, chemokines, growth factors, and proteases that impact tumour, immune, inflammatory, and other stromal cells. Even though, the composition of SASP varies depending on the cell and tissue of origin, the inflammatory cytokines IL-6 and IL-8 are consistently present and are responsible for maintaining and propagating the SASP response in the tumour microenvironment [50].

In our cell model, we observed that NR6 treatment induced *IL6*, but on the contrary, significantly inhibited *CXCL8* expression and IL-8 release. The observed differential expression of these two cytokines deserves further investigation. However, as IL-8 has been described to be the major attractant for neutrophils [51], we analysed the capability of MPM MCSs to recruit these immune cells. Neutrophils are the most abundant leukocytes circulating in human blood rapidly recruited to sites of tissue injury. It has long been assumed they have a short half-life and are rapidly cleaned from circulation [52]. However, recent research has challenged old paradigms, and neutrophils have gained increased attention. Recent reviews have provided evidence of their function in cancer progression [53]. Tumour-associated neutrophils (TAN) show a high level of plasticity and can exert dual functions. TANs can be part of tumour-promoting inflammation or conversely, mediate antitumour responses [54]. A key function of neutrophils is their ability to influence the behaviours of other immune cells. The complexity of neutrophil-based immunosuppressive mechanisms was recently exemplified by evidence that neutrophil extracellular traps (NETs) released in the tumour microenvironment shield cancer cells from cytotoxic immune cells [55]. Here we describe that MPM MCSs recruited neutrophils and induced NETosis. NR6 treatment prevented neutrophil recruitment by exerting its effect on tumour cells, as neutrophils did not express ALDH1A3 and were insensitive to its inhibition. The phenotype and the effects of neutrophils recruited inside MCSs need to be further characterized.

Another open question is whether senescence and apoptosis are truly alternative cell fates. One hypothesis is that cellular changes that are pro-senescent are actively anti-apoptotic and that senescent cells are resistant to apoptosis [56]. The critical role for p16^ink4a^ in cell-fate determination, following genotoxic stress, has been extensively discussed [57]. Here we describe that *CDKN2A* plays a key role in MPM response to NR6 treatment. NR6 induced senescence in *CDKN2A* proficient cells, while induced apoptotic death in *CDKN2A* silenced or homozygously deleted cells. Therefore, one-two punch approaches combining NR6 with senolytics to remove *CDKN2A*-positive senescent cells deserve to be explored. Deletion of *CDKN2A* is a common molecular alteration in MPM, and is associated with shorter patient survival [58]. In addition to abemaciclib as a treatment option for patients with p16^ink4a^ negative relapsed MPM [10], we identified a novel therapeutically exploitable vulnerability of *CDKN2A* null MPM cells.

This study highlights that targeting ALDH1A3, via selective pharmacological inhibition, is effective in MPM cell models. These results should be further validated using a 3D culture models, in which cancer cells form spheroids within matrices as an attempt to better mimic the *in vivo* microenvironment [59,60].

In conclusion, our findings present ALDH1A3 as an attractive target for the therapeutic management of MPM.

## 4. Materials and Methods

### 4.1. Reagents and Antibodies

The monoclonal antibodies specific for Poly (ADP-ribose) polymerase1 (PARP-1) and α-Tubulin were purchased from Santa Cruz Biotechnology (Santa Cruz, CA, USA). The polyclonal antibody specific for ALDH1A3 was purchased from Abcam (Cambridge, UK). The polyclonal antibodies specific for histone H2AX and gamma histone 2AX (γ-H2AX) were purchased from Cell Signaling Technology (Danvers, MA, USA). Anti-mouse and anti-rabbit IgG peroxidase conjugated antibodies and reagents were from Sigma-Aldrich (St. Louis, MO, USA). The monoclonal antibody for human CD66b, the FITC conjugated goat anti-mouse and the FITC conjugated goat anti-mouse were from Invitrogen-Thermo Fisher (Waltham, MA, USA). The monoclonal antibody anti human MPO and the PE conjugated goat anti-mouse were from eBiosciences-ThermoFisher, while the polyclonal antibody anti human Cit-Histone H3 (Arg2, Arg8, Arg17) was from Abbomax (San Jose, CA, USA). Nitrocellulose membrane and ECL were bought from Bio-Rad (Hercules, CA, USA). Lipofectamine transfection reagent, sera, culture medium, and antibiotics were from ThermoFisher (Waltham, MA, USA). Non-specific (NS) or specific *CDKN2A* siRNAs were from Qiagen (Hilden, Germany). The highly selective and potent ALDH1A3 inhibitor NR6 was synthesized and characterized as previously described [23]. The molecular structure of NR6 is reported in Gelardi et al. [23].

### 4.2. Cell Cultures and Transfection

The biphasic MPM derived MSTO-211H cell line was obtained from the Istituto Scientifico Tumori (IST) Cell-bank, Genoa, Italy; the epithelioid MPM derived REN cell line was isolated, characterized and kindly provided by Dr. Albelda S.M. (University of Pennsylvania, Philadephia, PA, USA); the H2596 cell line, isolated by Dr. Pass H.J. from surgical specimens derived from patients with resected sarcomatoid MPM, was kindly provided by Dr. Thomas W. (RCSI, Dublin, Ireland). The mesothelial NP2 cells were kindly provided by Dr. Steven Gray (Trinity College, Dublin, Ireland). Cells were cultured in standard conditions in RPMI-1640 medium supplemented with 10% FBS, 1% L-glutamine and 1% penicillin-streptomycin at 37 °C in humidified incubator with 5% CO_2_. Cells grown to 80% confluence in tissue culture dishes were transiently transfected with non-specific (NS) or specific *CDKN2A* siRNAs using Lipofectamine reagent. To obtain cell number and viability information after treatments, cells were trypsinized, stained with Trypan blue, and counted in a Bürker chamber.

### 4.3. Multicellular Spheroids (MCSs)

MCSs were generated according to a published protocol [61] of 1% agarose in sterile water solution. Before use, the coated plates were sterilized by UV light for 20 min. In each well 1 × 10^4^ cells were seeded in 100 μL of RPMI 1640 medium, supplemented with 1% L-Glutamine and 1% penicillin-streptomycin and 10% of FBS. After an overnight culture in humidified incubator, at 37 °C, 5% CO_2_, cells aggregate to form a single MCS/well. MCSs were kept in culture for a maximum of 72 h.

### 4.4. Malondialdehyde Quantitation

About 50 µL of a 5,5-dimethylcyclohexane-1,3-dione (DCHD) 20 mM working solution (DCHD dissolved in 2.5 g of ammonium acetate trihydrate and 2.5 mL of glacial acetic acid in 25 mL of deionized water) were added to 50 µL of spheroids lysate samples and the mixture was incubated in a 60 °C water bath for 1 h. Samples were diluted by adding 200 µL of acetonitrile and centrifuged at 13,000 rpm for 10 min, than the supernatants were injected onto the LC system. Quantification of malondialdehyde in samples was performed by using the calibration curve obtained by analysis of malondialdehyde tetrabutylammonium standard solutions in the range 10–1000 nM derivatized with DCHD.



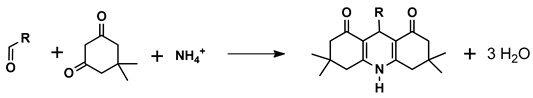



       Derivatization of aldehydes with DCHD.

LC-HRMS chromatogram, M2 spectrum of MDA derivatized with DCHD and additional information, are shown in Appendix A.

### 4.5. NAD^+^ Quantitation

For NAD^+^ quantitation, a pool of 10 MCSs was extracted with 400 μL of NADH/NAD^+^ Extraction Buffer by two freeze/thaw cycles (20 min on dry ice, followed by 10 min at room temperature) and vortexed for 10 s. After centrifugation in a cold microcentrifuge 5 min at 4 °C at top speed, supernatant (containing extracted NAD^+^/NADH) were collected and transferred into new tubes and then deproteinized by filtering the samples through a 10 kD Spin Column (ab93349). The levels of both NADt (total NAD^+^ and NADH) and NADH were measured using the ab65348 NAD^+^/NADH Assay Kit (Abcam, Cambridge, UK) according to the manufacturer’s instructions. After reading absorbance at OD 450 nm, the level of NAD^+^ was calculated by subtracting NADH from NADt and normalized on cell number.

### 4.6. IL-8 Measurement

IL-8 was quantified using the commercially available kit anti-human IL-8 ELISA MAX Delux set (BioLegend Global Headquarters, San Diego, CA, USA) according to the manufacturer’s instructions. The expected minimum detectable concentration of IL-8 for this set is 8 pg/mL.

### 4.7. Isolation of Human Neutrophils

Human neutrophils were collected and isolated from venous blood samples from healthy volunteers. An amount of 20 mL of venous blood was mixed with 10 mL of 0.9% saline with Dextran 500, and the left standing for 30 min at room temperature to allow sedimentation of red cells. After the sedimentation, leukocyte-rich supernatant was recovered and centrifuged at 1200 rpm for 10 min. The pellet was diluted in 8 mL of PBS and carefully stratified over 4 mL Ficoll-Paque Plus and then centrifugated at 1800 rpm for 15 min. The supernatant, which contains a mononuclear cell layer, was discarded. For lysing red blood cells, the remaining pellet was resuspended with 0.2% NaCl for 30 s and then mixed with an equal volume of 1.6% NaCl. The neutrophils were washed, pelleted, and resuspended in RPMI-1640 supplemented with 100 IU/mL penicillin, 0.1 mg/mL streptomycin and 0.25 µg/mL amphotericin B. The percentage of neutrophils was evaluated by cytofluorimetric analysis using anti CD14 (monocytes), CD3 (T lymphocytes), and CD66b (neutrophils) antibodies and was considered satisfactory when CD66b > 90%. For co-cultures experiments, 4 × 10^4^ neutrophils were added to each MCS, previously treated with NR6 (1 µM) or DMSO for 48 h, and incubated at 37 °C in 5% CO_2_ for an additional 24 h.

### 4.8. Confocal Microscopy Analysis

MCSs were let to adhere for 1 h to poly-L-lysine coated glass cover slips and then fixed in 4% paraformaldehyde (PFA) for 30 min, at room temperature. Cells were permeabilized with 0.5% Triton-X100 in PBS for 5 min, at room temperature. After fixation MCSs were rinsed in PBS and incubated blocking solution (10% FBS in PBS) for 1 h. Primary antibodies diluted (1:50) in 2% FBS in PBS were incubated at room temperature for 1 h. After washing with 2% FBS in PBS the immunoreactivity was revealed using the secondary antibodies diluted (1:100) in 2% FBS in PBS for 30 min, at room temperature. Negative controls were performed by substituting the primary antibodies with the 2% FBS in PBS buffer. The immunostained MCSs were counterstained with Diamidino-2-phenylindole (DAPI), rinsed with PBS and mounted on slides using fluorescent aqueous mounting medium (Agilent Dako, Santa Clara, CA, USA). Confocal imaging was performed using a LSM700 laser-scanning confocal microscope (Carl Zeiss, Le Pica, France) with 63× magnification. To image the entire spheroid laser scanning microscope acquired single plain tile scans that have been automatically stitched into a larger mosaic.

### 4.9. Protein Extraction and Immunoblot

Cells were extracted with 1% NP-40 lysis buffer (50 mM Tris-HCl pH 8.5 containing 1% NP-40, 150 mM NaCl, 10 mM EDTA, 10 mM NaF, 10 mM Na_4_P_2_O_7_, and 0.4 mM Na_3_VO_4_) with freshly added protease inhibitors (10 μg/mL leupeptin, 4 μg/mL pepstatin, and 0.1 Unit/mL aprotinin). Lysates were centrifuged at 13,000 rpm for 10 min at 4 °C and the supernatants were collected and assayed for protein concentration with the Bradford assay method (Bio-Rad). For histones analysis 5 × 10^5^ cells were lysed in a 4× pellet volume of Buffer A (300 mM sucrose, 10 mM HEPES, 10 mM KCl, 2 mM MgCl_2_, and 1 mM EGTA) supplemented with 0.15% NP-40, protease, and phosphatase inhibitors. Cells were then centrifuged at 1300 rpm for 5 min at 4 °C. The pellet (nuclei) was washed 5 times with Buffer B (50 mM HEPES, 0.4 M NaCl, 1 mM EDTA) and then resuspended in 3× pellet volume of Buffer B supplemented with protease inhibitors, sonicated, and then incubated on a thermomixer for 20 min at 4 °C with 1300 rpm. Then samples were centrifuged for 15 min at 13,000 rpm at 4 °C, and the supernatant containing the nuclear proteins was quantified and used for downstream applications. Proteins were separated by SDS-PAGE under reducing conditions. Following SDS-PAGE, proteins were transferred to nitrocellulose, reacted with specific antibodies, and then detected with peroxidase-conjugate secondary antibodies and chemioluminescent ECL reagent. Digital images were taken with the Bio-Rad ChemiDocTM Touch Imaging System and quantified using Bio-Rad Image Lab 5.2.1.

### 4.10. RNA Isolation and Real-Time PCR

Total RNA was extracted using the guanidinium thiocyanate method. Starting from equal amounts of RNA, cDNA used as template for amplification in the real-time PCR (5 µG), was synthesized by the reverse transcription reaction using RevertAid Minus First Strand cDNA Synthesis Kit from Fermentas-Thermo Scientific (Burlington, ON, Canada), using random hexamers as primers, according to the manufacturer’s instructions. An amount of 20 ng of cDNA was used to perform RT-PCR amplification of mRNA. The real-time reverse transcription-PCR was performed using the double-stranded DNA-binding dye SYBR Green PCR Master Mix (Fermentas-Thermo Scientific, Burlington, ON, Canada) on an ABI GeneAmp 7000 Sequence Detection System machine, as described by the manufacturer. The instrument, for each gene tested, obtained graphical Cycle threshold (Ct) values automatically. Triplicate reactions were performed for each marker and the melting curves were constructed using Dissociation Curves Software (Applied Biosystems, Foster City, CA, USA), to ensure that only a single product was amplified.

### 4.11. Statistical Analysis

Statistical evaluation of the differential analyses was performed by one-way ANOVA and Student’s *t*-test.

## Figures and Tables

**Figure 1 ijms-24-06689-f001:**
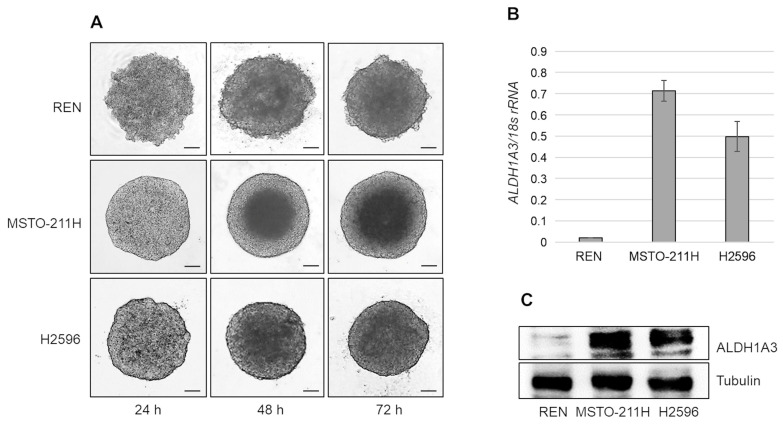
ALDH1A3 expression in MPM cell lines derived MCSs. (**A**) Representative phase contrast images (×40 magnification) of REN, MSTO-211H, and H2596 cells cultured as MCSs for 24, 48, and 72 h. Scale bar = 100 µm. (**B**) Bar graph shows *ALDH1A3* mRNA expression evaluated by real time-PCR in REN, MSTO-211H, and H2596 MCSs, at 72 h. Data are expressed as *ALDH1A3* mRNA/*18S* rRNA ratio. Each bar represents mean ± s.d. of three independent experiments. (**C**) Representative Western blot analysis of ALDH1A3 expression in REN, MSTO and H2596 MCSs at 72 h. Tubulin was used as loading control.

**Figure 2 ijms-24-06689-f002:**
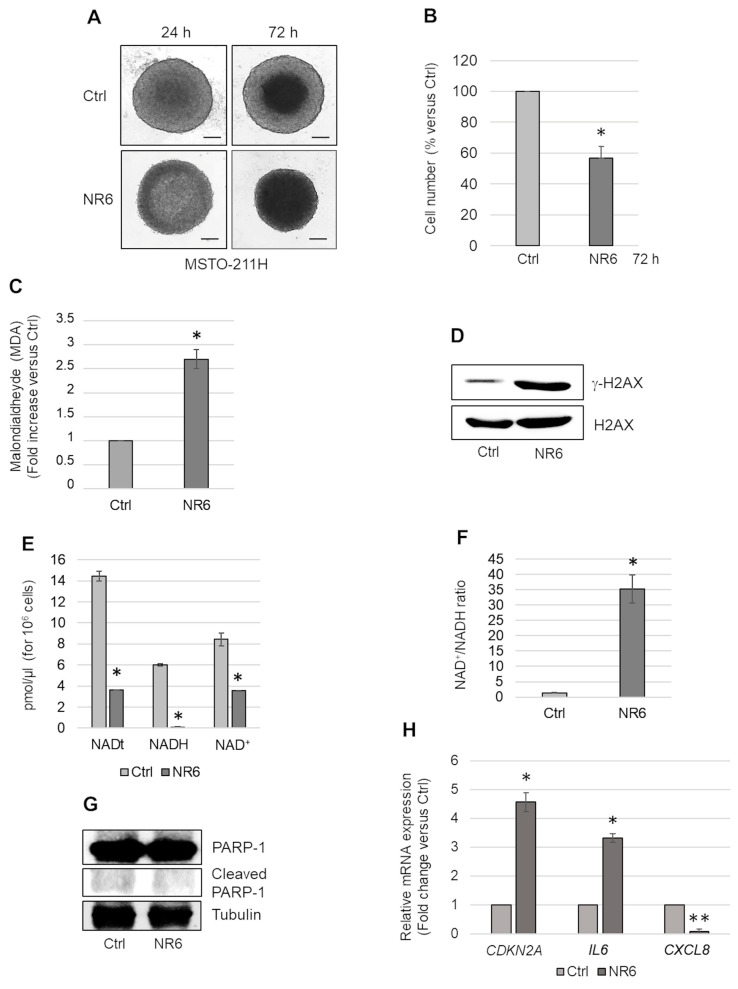
Effects of NR6 treatment of MSTO-211H MCSs. (**A**) Representative phase contrast images (×40 magnification) of MSTO-211H cultured as MCSs and treated with NR6 for 24 or 72 h. Scale bar = 100 µM. (**B**) Bar graph shows the number of viable cells in MSTO-211H MCSs treated, 72 h, with NR6 represented as percentage versus untreated control MCSs (Ctrl). (**C**) Bar graph shows the intracellular level of malondialdheyde (MDA) in MSTO-211H MCSs treated 72 h with NR6, expressed as fold increase versus untreated control MCSs (Ctrl). (**D**) Representative Western blot analysis of γH2AX expression in MSTO-211H MCSs treated or not with NR6, for 72 h. H2AX was used as loading control. (**E**) Bar graph shows the levels of NADt, NADH, and NAD^+^ in MSTO-211H MCSs untreated (Ctrl) or treated with NR6, 72 h. (**F**) Bar graph shows the percentage of NAD^+^/NADH in untreated (Ctrl) MSTO-211H MCSs or treated with NR6, 72 h. (**G**) Representative Western blot analysis of PARP-1 expression/cleavage in MSTO-211H MCSs treated or not with NR6, for 72 h. Tubulin was used as loading control. (**H**) Bar graph shows *CDKN2A*, *IL6* and *CXCL8* mRNA expression evaluated by real time-PCR in MSTO-211H MCSs treated with NR6, for 72 h. Data are expressed as fold change versus untreated control MCSs (Ctrl). *18S* rRNA was used as housekeeping gene. In all graphs reported in Figure 2, each bar represents mean of three independent experiments ± s.d., * *p* ≤ 0.05, ** *p* ≤ 0.01.

**Figure 3 ijms-24-06689-f003:**
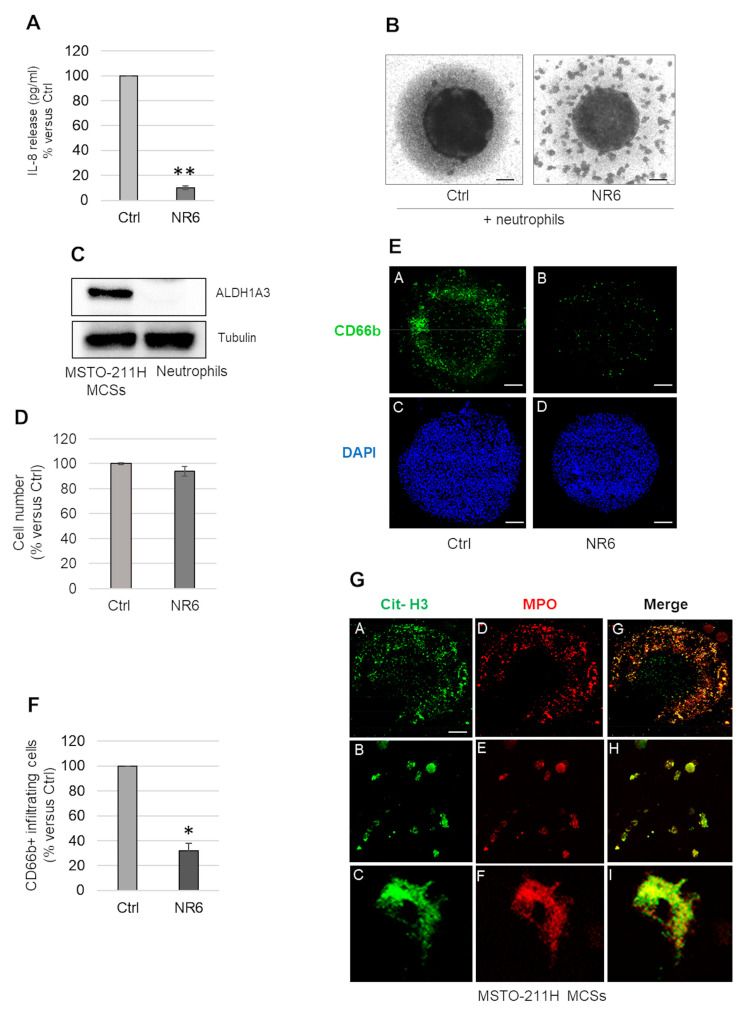
Effects of NR6 treatment on neutrophil recruitment ability of MSTO-211H MCSs. (**A**) Bar graph shows the IL-8 level (pg/mL) released in the medium by NR6-treated (72 h) MSTO-211H MCSs expressed as percentage versus untreated control MCSs (Ctrl). (**B**) Representative phase contrast images of MSTO-211H MCSs treated or not with NR6, for 48 h, and then co-cultured with neutrophils for an additional 24 h. Scale bar = 100 µm. (**C**) Representative Western blot analysis of ALDH1A3 expression in neutrophils compared to MSTO-211H MCSs. Tubulin was used as loading control. (**D**) Bar graph shows the percentage of viable neutrophil upon 24 h of NR6 treatment, expressed as percentage versus untreated controls (Ctrl). (**E**) Representative confocal images of MSTO-211H MCSs treated or not, 48 h, with NR6 and co-cultured with neutrophils for additional 24 h. Neutrophils were stained with anti-CD66b-FITC antibodies (green) (**A**,**B**), nuclei were counterstained with DAPI (blue) (**C**,**D**). Scale bar = 200 µm (**F**). Bar graph shows the percentage versus control of neutrophils infiltrated in MSTO-211H MCSs treated with NR6, evaluated by anti-CD66b-FITC antibodies staining and FACS analysis. (**G**) Representative confocal images of neutrophils infiltrated in a MSTO-211H MCS, stained with anti-Cit-H3-FITC (green) (**A**–**C**) and anti-MPO-PE (red) (**D**–**F**) antibodies at 63× magnification (**A**,**D**,**G**). Images were merged (**G**,**H**,**I**) and zoomed (**B**,**C**,**E**,**F**,**H**,**I**). Scale bar = 200 µm. In all graphs reported in Figure 3, each bar represents mean of three independent experiments ± s.d., * *p* ≤ 0.05, ** *p* ≤ 0.01.

**Figure 4 ijms-24-06689-f004:**
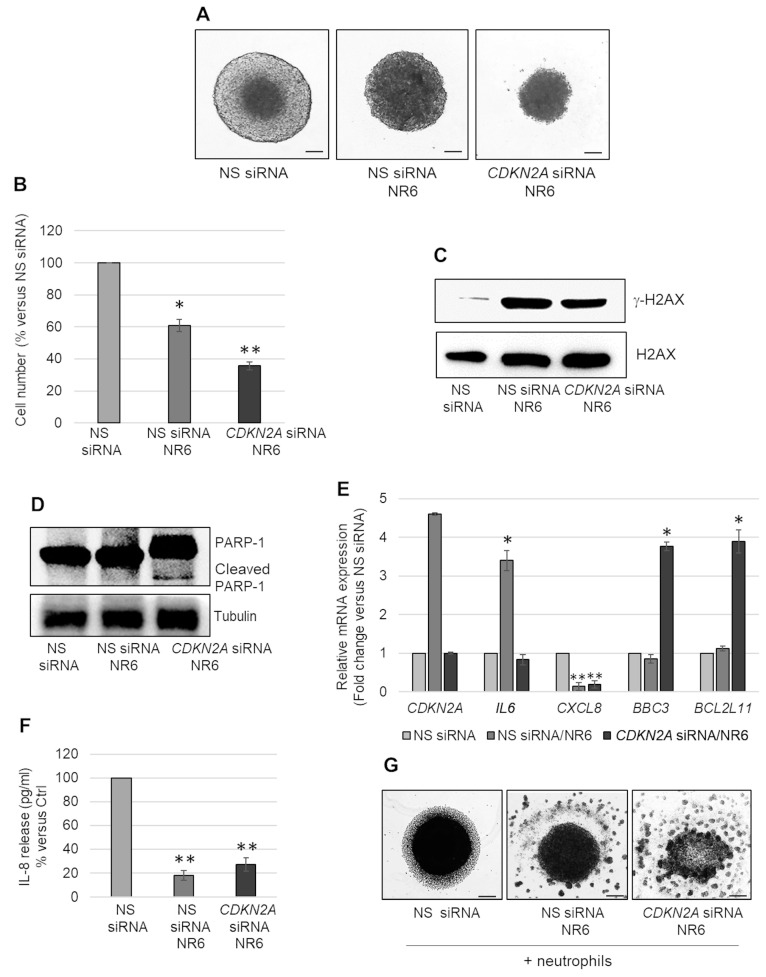
Effects of NR6 treatment in *CDKN2A* silenced MSTO-211H MCSs. (**A**) Representative phase contrast images of MSTO-211H MCSs transfected with non-specific siRNA (NS siRNA) or specific *CDKN2A* siRNA (*CDKN2A* siRNA) treated or not with NR6, for 72 h. Scale bar = 100 µm. (**B**) Bar graph shows the number of viable cells in NS siRNA and in *CDKN2A* siRNA MCSs treated or not with NR6, for 72 h, represented as percentage versus untreated NS siRNA. (**C**) Representative Western blot analysis of γ-H2AX expression in NS siRNA and *CDKN2A* siRNA MCSs treated or not with NR6, for 72 h. H2AX was used as loading control. (**D**) Representative Western blot analysis of PARP-1 expression/cleavage in NS siRNA and *CDKN2A* siRNA MCSs treated or not with NR6, for 72 h. Tubulin was used as loading control. (**E**) Bar graph shows *CDKN2A*, *IL6*, *CXCL8*, *BBC3* and *BCL2L11* mRNA expression in NS siRNA and *CDKN2A* siRNA MCSs treated with NR6, for 72 h, expressed as fold changes versus untreated NS siRNA. *18S* rRNA was used as housekeeping gene. (**F**) Bar graph shows the levels (pg/mL) of IL-8 released in the medium by MSTO-211H MCSs transfected with NS siRNA or specific *CDKN2A* siRNA and treated with NR6, for 72 h. IL-8 levels are expressed as percentage versus untreated NS siRNA. (**G**) Representative phase contrast images of MSTO-211H MCSs transfected with NS siRNA or specific *CDKN2A* siRNA treated or not with NR6, for 48 h, and then co-cultured with neutrophils for additional 24 h. Scale bar = 100 µm. In all graphs reported in Figure 4, each bar represents mean of three independent experiments ± s.d., * *p* ≤ 0.05, ** *p* ≤ 0.01.

**Figure 5 ijms-24-06689-f005:**
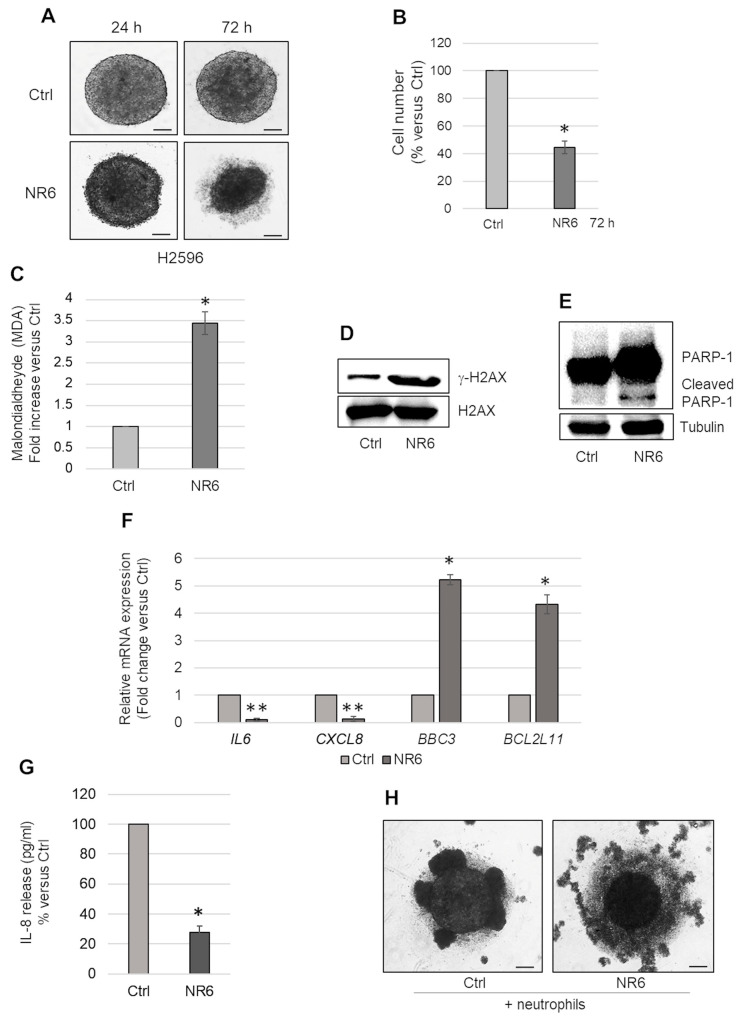
Effects of NR6 treatment in *CDKN2A* null H2596 MCSs. (**A**) Representative phase contrast images of H2596 MCSs treated with NR6, for 24 and 72 h. Scale bar = 100 µm. (**B**) Bar graph shows the number of viable cells in NR6-treated H2596 MCSs, represented as percentage versus untreated MCSs (Ctrl). (**C**) Bar graph shows the intracellular level of malondialdheyde (MDA) in NR6-treated H2596 MCSs (NR6) expressed as fold increase versus untreated MCSs (Ctrl). (**D**) Representative Western blot analysis of γ-H2AX expression in untreated (Ctrl) or NR6-treated H2596 MCSs (NR6). H2AX was used as loading control. (**E**) Representative Western blot analysis showing PARP-1 expression/cleavage in untreated (Ctrl) or NR6-treated H2596 MCSs. Tubulin was used as loading control. (**F**) Bar graph shows *CDKN2A*, *IL6*, *CXCL8*, *BCL2L11,* and *BBC3* mRNA expression in NR6-treated H2596 MCSs (NR6) evaluated by real time-PCR and expressed as fold changes versus untreated control MCSs (Ctrl). *18S* rRNA was used as housekeeping gene. (**G**) Bar graph shows the levels (pg/mL) of IL-8 released in the medium by H2596 MCSs treated with NR6, expressed as percentage versus untreated MCSs (Ctrl). (**H**) Representative phase contrast images of H2596 MCSs treated or not with NR6, for 48 h, and then co-cultured with neutrophils for additional 24 h. Scale bar = 100 µm. In all graphs reported in Figure 5, each bar represents mean of three independent experiments ± s.d., * *p* ≤ 0.05, ** *p* ≤ 0.01.

## Data Availability

The data presented in this study are included in this publication article and its additional files. All data can be shared upon request by email.

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
