# Peer review of "A Selective ALDH1A3 Inhibitor Impairs Mesothelioma 3-D Multicellular Spheroid Growth and Neutrophil Recruitment"

_ijms, 2023, doi:10.3390/ijms24076689_

Round 1

Reviewer 1 Report

This is a well written manuscript.  A few minor recommendations:

1) If possible, the chemical structure of NR6 could be added as a figure.

2) Reference 59 could better be cited in the Introduction after the sentence in line 52.  The work with abemiciclib confirms the importance of CDKN2a deletion as a therapeutic target in mesothelioma.

Author Response

Dear referee,

We would like to thank you for the time and the efforts you have spent for the evaluation of our manuscript. Below you can find our point-by-point reply to your fruitful comments (Track Changes in the new uploaded version of the manuscript).

1) The molecular structure of NR6 has been previously reported by Gelardi et al. (reference number 23). In the material and methods section we have added a sentence to clarify this point, with the relative reference 23.

2) Thanks for the observation. As suggested, to underline the importance of CDKN2A deletion as a therapeutic target in mesothelioma, we have introduced the reference 59, now listed as reference 10, in the introduction section.

3) In the revised version minor spell changes have been introduced.

Reviewer 2 Report

The study is based on spheroids but there is no extracellular matrix or tumor microenvironment matrix embedded in spheroids. In the discussion and future part, please add a section that these results should be further validated using 3D culture models where the cancer cells will form spheroids in such matrices which will further suit the human microenvironment-mimicking matrices. (https://doi.org/10.1016/j.yexcr.2018.06.037 ; https://doi.org/10.1186/s12885-015-1944-z ). I recommend using the references as the data was analyzed using animal and human-microenvironment-based 3D tissue models. 

Author Response

Dear referee,

We would like to thank you for the time and the efforts you have spent for the evaluation of our manuscript. Below you can find our reply to your fruitful comments (Track Changes in the new uploaded version of the manuscript).

  • Thanks for the suggestion. As suggested, in the discussion section we have added a sentence to clarify that our results should be further validated using 3D culture models in which cancer cells form spheroids within matrices  as an attempt to better mimic the in vivo microenvironment. We have added the suggested references, listed as references 60 and 61.
  • In the revised version minor spell changes have been introduced.